# Predicting the Loose Zone of Roadway Surrounding Rock Using Wavelet Relevance Vector Machine

**Yang Liu [1,2], Yicheng Ye [1,3], Qihu Wang [1,\*], Xiaoyun Liu [1] and Weiqi Wang [1]**

[1] Resources and Environmental Engineering Institute, Wuhan University of Science and Technology, Wuhan 430081, China; liuyang@wust.edu.cn (Y.L.); yeyicheng@wust.edu.cn (Y.Y.); liuxiaoyun@wust.edu.cn (X.L.); Wangweiqi@wust.edu.cn (W.W.)

[2] Hubei Key Laboratory of Mechanical Transmission and Manufacturing Engineering, Wuhan University of Science and Technology, Wuhan 430081, China

[3] Industrial Safety Engineering Technology Research Center of Hubei Province, Wuhan 430081, China

\* Correspondence: wangqihu@wust.edu.cn; Tel.: +86-027-6886-2885

**Abstract:** By applying the Wavelet Relevance Vector Machine (WRVM) method, this research proposes the loose zone of roadway surrounding rock prediction. Based on the theory of relevance vector machine (RVM), the wavelet function is introduced to replace the original Gauss function as the model kernel function to form the WRVM. Five factors affecting the loose zone of roadway surrounding rock are selected as the model input, and the prediction model of the loose zone of roadway surrounding rock based on WRVM is established. By using cross-validation method, the kernel parameters of three kinds of wavelet relevance vector machines (RVMs) are calculated. By comparing and analyzing the root mean square (RMS) error of the test results of each predictive model, the advantages and accuracy of the model are verified. In practical engineering applications, the average relative prediction errors of the Mexican relevance vector machine, the Morlet relevance vector machine and the difference of Gaussian (DOG) relevance vector machine models are accordingly 4.581%, 4.586% and 4.575%. The square correlation coefficient of the predicted samples is 0.95 > 0.9, which further verifies the accuracy and reliability of the proposed method.

**Keywords:** relevant vector machine (RVM); wavelet relevance vector machine (WRVM); wavelet kernel function; loose zone of roadway surrounding rock; prediction model

## 1. Introduction

A surrounding rock loose zone refers to the stress redistribution of surrounding rock after excavation and the stress variation in surrounding rock, which leads to stress concentration. When the stress value exceeds the strength limit or yield limit, the roadway surrounding rock mass breaks up and forms a certain range of rupture area [1]. For many years, rock bolts are commonly used to enhance the stability of surrounding rock [2–5]. However, rock bolts are subjected to the loose zone of pre-load [6–8] or corrosion [9–11] and may fail [12]. The thickness determination of loose zone is not only a significant factor for the roadway surrounding rock's stability, but also an important basis for the design of roadway support. Therefore, the accurate determination for the thickness of loose zone of roadway surrounding rock makes a great difference to improve the stability of underground engineering rock mass [13–16].

With the development of structural health monitoring [17–19], the methods to determine the loose zone thickness of roadway surrounding rock generally include acoustic wave test, numerical simulation and, support vector machine prediction [20,21]. The acoustic wave test method [22] can directly and accurately determine the range of loose zone, however the field operation is cumbersome

and the cost is high. The numerical simulation method [23] is low cost and easy to realize. Nevertheless, it is obviously affected by the parameters of surrounding rock, and the result sometime has a large margin of error. Due to the nonlinearity of loose zone for surrounding rock and uncertainty of state and parameter measurement, the prediction of surrounding rock loose zone can be regarded as a typical complex and fuzziness problem. In recent decades, various intelligent methods, such as artificial neural network (ANN) [24–26], fuzzy logic [27–29], genetic algorithm (GA) [30,31], wavelet packet analysis (WPA) [32–34], supporting vector machine (SVM) [35–37], among others [38–40], have been developed to deal with nonlinearities and uncertainties. The support vector machine prediction method [41] can solve the problems of nonlinearity and small samples, and has achieved some success in the prediction of roadway surrounding rock loose zone. Nevertheless, the prediction accuracy and generalization ability of the model are affected by parameters.

Relevant vector machine (RVM) [42–46] is a sparse probability model alike support vector machine (SVM) proposed by Tipping in 2001. Its training process is based on Bayesian framework [47,48]. RVM can be used for regression estimation and prediction to obtain the distribution of predicted values. It not only has excellent nonlinear fitting and generalization ability, but also makes up for the defect that SVM needs to estimate regularization parameters and cannot get the prediction results based on probability [49,50]. RVM combines the advantages of core-based method and Bayesian theory, and establishes the relationship between a group of input vectors and its required output. Therefore, the selection of kernel functions plays an important role in RVM achieving good performance.

In practical applications, the commonly used kernels are Gauss kernels and polynomial kernels [51], however they cannot generate a complete set of bases in a space (such as square integrable space) by translation. The incomplete base will result in RVM not approaching any objective function in the kernel space. Wavelet is a function approximation tool in square integrable space [52,53]. Wavelet kernel function has been applied to SVM function approximation, which effectively improves the accuracy of the model [51]. Wavelet technology has great potential in dealing with approximation and classification of non-stationary signals [54,55]. Wavelet function is a group of bases [56], which can approximate any function. Therefore, this paper has combined wavelet technology with RVM, namely wavelet relevant vector machine (WRVM), and applied it to prediction of surrounding rock loosening zone of roadway. The feasibility and effectiveness of this method can be verified by practical engineering cases.

The paper is structured as follows: Section 2 shows the conception of the RVM and WRVM and the reasoning process of WRVM. Section 3 presents the key steps of the prediction model of surrounding rock loose zone using wavelet relevance vector machine. Section 4 compares the accuracy of the prediction models of SVM, RVM and WRVM, and predicts the loose zone of surrounding rock of 15 roadways by using the prediction model of WRVM. Section 5 concludes the paper.

## 2. Wavelet Relevance Vector Machine

RVM can be used for regression estimation and prediction to obtain the distribution of predicted values. It not only has excellent nonlinear fitting and generalization ability, but also makes up for the defect that SVM needs to estimate regularization parameters and cannot get the prediction results based on probability.

### 2.1. Relevant Vector Machine

The training sampling test is given by $\{x_i, t_i\}_{i=1}^n, x_i \in R^d, t_i \in R$. The RVM model output is defined as [57],

$$y(x, \omega) = \sum_{i=1}^n \omega_i K(x, x_i) + \omega_0 \tag{1}$$

In the formula, $n$ is the number of sample, $\omega_i$ the weight of the model, $K(x, x_i)$ the kernel function, the sample matching non-zero $\omega_i$ is called the relevance vector. The Figure 1 illustrates the RVM architecture model.

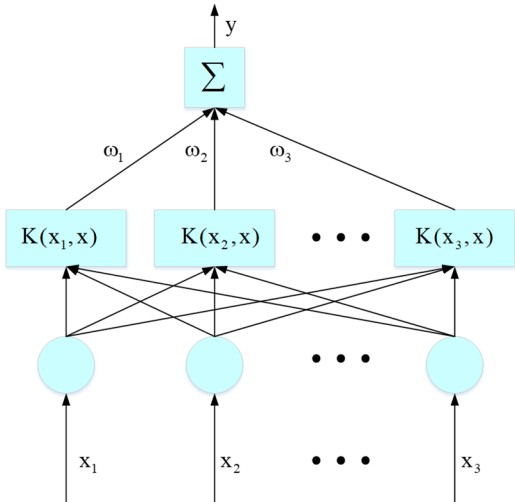

**Figure 1.** The architecture of a relevant vector machine (RVM).

RVM inherits SVM modeling using structural risk minimization which has good generalization capability. However, SVM cannot give the confidence range. In order to make up for the defect that SVM cannot get the prediction results based on probability, RVM needs to learn the model in the Bayesian framework.

Assuming that the training samples are independent and contain Gauss noise with variance $\sigma^2$, the training sample' likelihood function is expressed in the following [58–60]:

$$p\left(t|\omega, \sigma^2\right) = \prod_{i=1}^{n} p\left(t_i|\omega, \sigma^2\right) = \left(2\pi\sigma^2\right)^{-n/2} \exp\left\{-\frac{1}{2\sigma^2} t - \Phi\omega^2\right\} \tag{2}$$

where

$$t = (t_1, \cdots, t_n)^T, \omega = (\omega_0, \cdots, \omega_n)^T, \Phi_{n \times (n+1)} = [\phi(x_1), \phi(x_2), \cdots \phi(x_n)]^T$$
$$\phi(x_i) = \phi_i = [1, K(x_i, x_1), \ldots, K(x_i, x_n)]^T.$$

According to the structural risk minimization, if the weights are not constrained, the direct maximization Equation (2) will cause serious over-fitting. To enhance the model's generalization capability, the definition of the distribution of Gaussian prior probability for each weight value is made by RVM [60,61],

$$p(\omega|\alpha) = \prod_{i=0}^{n} N\left(0, \alpha_i^{-1}\right) \tag{3}$$

where $\alpha$ is a parameter determining the prior distribution of weights $\omega$.

According to the prior probability and likelihood distribution, the distribution of posterior probability by calculating the weights using Bayesian criterion is as follows [48,59],

$$p\left(\omega|t, \alpha, \sigma^2\right) = \frac{p\left(t|\omega, \sigma^2\right) p(\omega|\alpha)}{p\left(t|\alpha, \sigma^2\right)} = N\left(\mu, \sum\right) \tag{4}$$

where the mean value $\mu = \sigma^{-2} \sum \Phi^T t$, the covariance $\mu = \sigma^{-2} \sum \Phi^T t$, $A = diag(\alpha_0, \alpha_1, \cdots, \alpha_l)$, when $\alpha \to \infty, \mu_i = 0$.

Therefore, the weights estimation is decided by the mean $\mu$ of the weight's posterior distribution, and the uncertainty of optimal weight value $\Sigma$ can be applied to express the uncertainty of model prediction. To evaluate the model weights, the optimal value of the super-parameter must be estimated first. Based on the Bayesian framework, the likelihood distribution of hyper parameters is calculated as follows,

$$p\left(t|\alpha,\sigma^2\right) = \int p\left(t|\omega,\sigma^2\right)p(\omega|\alpha)d\omega = N(0,C) \tag{5}$$

where the covariance $C = \sigma^2 I + \Phi A^{-1} \Phi^T$.

The marginal likelihood derivative of log $\alpha_i$ is set to 0, the $\alpha_i$ updating formula is shown in the following:

$$\alpha_i^{new} = \frac{\gamma_i}{\mu_i^2} \tag{6}$$

where $\gamma_i = 1 - \alpha_i \sum_{ii}$, $\sum_{ii}$ refers to the diagonal element of a matrix $\Sigma$. The achievement of sparsity is made because most parameters $\alpha_i$ are evaluated as quite large values, so the basic functions are accordingly pruned by enforcing their weights to 0. During the progression of optimization, the vectors associated with the residual non-zero weights from the training set are called relevance vectors (RV) [48].

In addition, the marginal likelihood derivative of log $\sigma^2$ is set to zero, the $\sigma^2$ update formula is given:

$$\left(\sigma^2\right)^{new} = \frac{t - \Phi\mu^2}{n - \sum\limits_{i=0}^{n} \gamma_i} \tag{7}$$

By maximizing the hyper parametric likelihood distribution, its optimal value $\alpha_{MP}$, $\sigma^2_{MP}$ can be found. For the new observed data $x_*$ prediction is applied to calculate the predictive distribution:

$$p\left(t_*|t,\alpha_{MP},\sigma^2_{MP}\right) = \int p\left(t_*|\omega,\sigma^2_{MP}\right)p\left(\omega|t,\alpha_{MP},\sigma^2_{MP}\right)d\omega = N\left(\mu_*,\sigma^2_*\right) \tag{8}$$

where, $\mu_* = \mu^T \phi(x_*)$ is the model predictive value, $\sigma^2_* = \sigma^2 MP + \phi(x_*)^T \sum \phi(x_*)$ the predictive model's variance information, and the predictive interval can be given by $[\mu_* - \sigma_*, \mu_* + \sigma_*]$.

## 2.2. Wavelet Kernel Function

RVM transforms the input space into a high-dimensional space through the pre-selected kernel function while realizing data linearization in this space. In square integrable space, besides the inner product kernel form $K(x,x') = K(\langle x,x' \rangle)$, there are translation invariant kernels $K(x,x') = K(x-x')$ for kernels $K(x,x')$.

Assuming that $\varphi(x)$ is a wavelet generating function, the translation invariant form of the wavelet kernel function can be constructed as [51,54]:

$$K(x,x') = \prod_{i=1}^{d} \varphi\left(\frac{x_i - x'_i}{a}\right) \tag{9}$$

In the formula, $d$ is the input vector dimension, and $a$ refers to the scaling factor of the wavelet kernel function which is a constant that needs to be optimized. In the following section, several commonly used wavelet generating functions and their kernels are given.

(1) The Mexican hat wavelet generating function is $\varphi(x) = \left(1 - x^2\right)\exp\left(-\frac{x^2}{2}\right)$, and the corresponding Mexican hat wavelet kernel function is given:

$$K(x, x') = \prod_{i=1}^{d}\left[1 - \frac{(x_i - x'_i)^2}{a^2}\right]\exp\left[-\frac{(x_i - x'_i)^2}{2a^2}\right] \tag{10}$$

(2) The Morlet wavelet generating function is $\varphi(x) = \cos(1.75x) \times \exp\left(-\frac{x^2}{2}\right)$, and the corresponding Morlet wavelet kernel function is:

$$K(x, x') = \prod_{i=1}^{d}\cos\left[1.75\frac{x_i - x'_i}{a}\right]\exp\left[-\frac{(x_i - x'_i)^2}{2a^2}\right] \tag{11}$$

(3) The DOG wavelet generating function is $\varphi(x) = \exp\left(-\frac{x^2}{2}\right) - \frac{1}{2}\exp\left(-\frac{x^2}{8}\right)$, and the matching DOG wavelet kernel function is:

$$K(x, x') = \prod_{i=1}^{d}\left\{\exp\left[-\frac{(x_i - x'_i)^2}{2a^2}\right] - \frac{1}{2}\exp\left[-\frac{(x_i - x'_i)^2}{8a^2}\right]\right\} \tag{12}$$

### 2.3. Wavelet Relevance Vector Machine

The architecture model of WRVM is similar to that of standard RVM. As illustrated in Figure 1, their major difference lies in the nonlinear mapping method and kernel function. A brief overview of the reasoning process of WRVM is given, as shown in Figure 2.

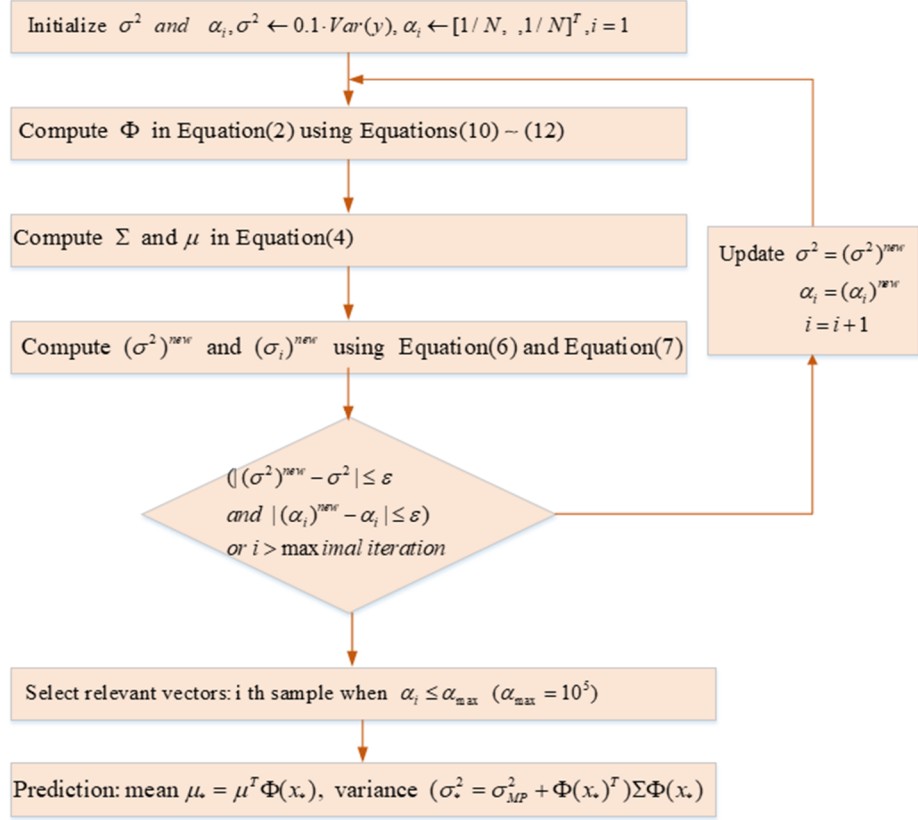

**Figure 2.** Wavelet relevance vector (WRV) machine algorithm procedure.

*2.4. Fitting Quality Estimation*

To compare the model's generalization capability, three measurements are introduced to test the capability of model fitting data and prediction [62], specifically as follows:

(1) Root mean square error (RMSE)

$$RMSE = \sqrt{\frac{\sum\limits_{i=1}^{n}(t_i - \mu_{*i})^2}{n}} \tag{13}$$

(2) Relative prediction error (RPE)

$$RPE = \frac{\sum\limits_{i=1}^{n}|t_i - \mu_{*i}|}{\sum\limits_{i=1}^{n}|t_i|} \tag{14}$$

(3) Square correlation coefficient $R^2$

$$R^2 = 1 - \frac{\text{SSR}}{\text{SSY}} \tag{15}$$

where SSR is the residual squares' sum, and SSY is the response variable squares' sum.

## 3. Establishment of Predictive Model for Loose Zone of Surrounding Rock in Roadway

According to the theory of WRVM, a predictive model of loose zone of surrounding rock in roadway using WRVM has been established. As shown in Figure 3, the concrete steps are:

(1) Determining input and output. Predicting the roadway surrounding rock loose zone, the main factors having an impact on the thickness of roadway surrounding rock loose zone are selected as the evaluation index, that is, the input of the prediction model (i = 1,2... D). The thickness of loose zone is the output of the prediction model Y.

(2) Preprocessing sample data. In order to eliminate the influence of inconsistent evaluation index dimension on data analysis, it needs to standardize the samples, that is, to divide the statistical data by the standard deviation of the data after zero-mean. It is to ensure that the mean value of each statistical data after processing is 0 and the variance is 1.

(3) Establishing the thickness prediction model of loose zone. The use of the cross-validation method [63] for the optimization of the kernel function parameters. The commonly used cross-validation methods include leave-out method, K-fold cross-validation and leave-one validation. Leave-out method was applied in this study to randomly separate the training samples into two parts. The first part is applied as training set to train the model. Another is used as validation set to validate the parameters and model. Therefore, the core parameters which render the smallest verification set RMS error can be selected. Using all training samples, the predictive model of roadway surrounding rock loose zone using WRVM has been established.

(4) Predicting loose zone thickness based on the model. Equation (6) can be used to predict the thickness of the loose zone after establishing the prediction model of the main factors and the loose zone thickness of roadway surrounding rock. Among them, $\mu_*$ is the model predictive value, $\sigma^2_*$ the variance information of the current predictive value.

(5) Analyzing model accuracy. After establishing the prediction model of surrounding rock loose zone, the accuracy of the model is evaluated by Equations (11)–(13).

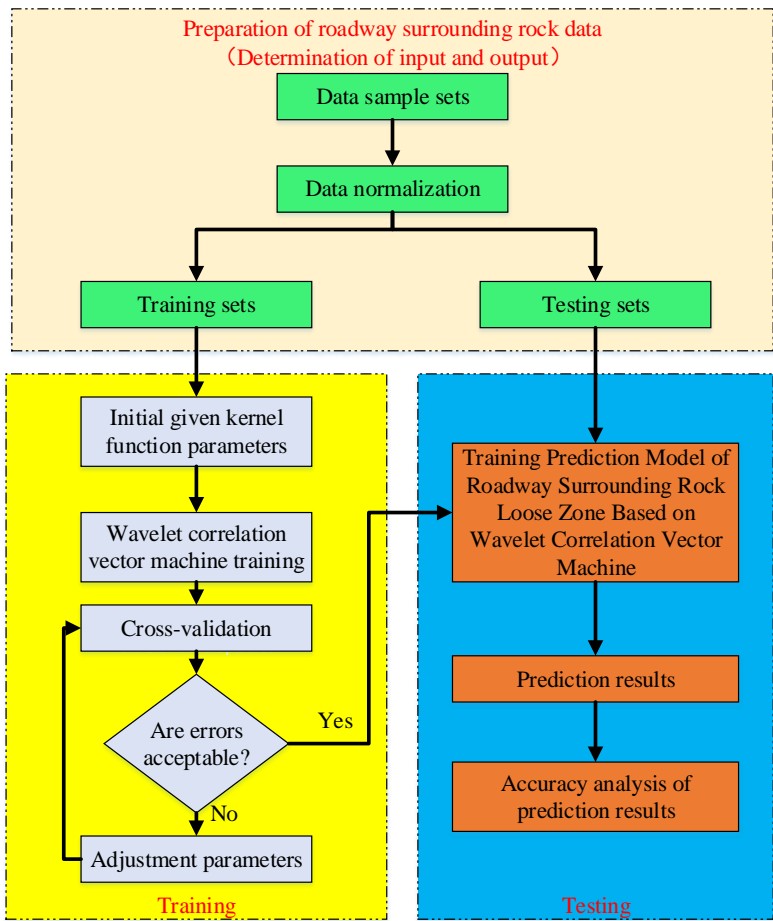

**Figure 3.** Prediction model of surrounding rock loose circle in roadway.

## 4. Application Analysis

### 4.1. Selection of Main Influencing Factors

There are many factors making a difference to the thickness of loose zone. The complex relationship between them often shows strong nonlinearity. Currently, there is no universal mathematical model. Based on the relevant literature [64,65] and the principles of simplicity, independence and accessibility, the influence of in situ stress on the size of loose zone is reflected by roadway burial depth, and the influence of roadway span and roadway section area on the size of loose zone is reflected by the geometric size and shape of roadway. Five indexes are used to predict the thickness of loose zone.

(1) Depth of roadway. The thickness of roadway surrounding rock loose zone increases with the roadway burial depth.

(2) Span of roadway. The thickness of roadway surrounding rock loose zone increases with the roadway span.

(3) Strength of surrounding rock. The difficulty of resisting failure of surrounding rock can be reflected by the surrounding rock strength, and the thickness of loose zone is inversely proportional to the surrounding rock strength. The strength of surrounding rock can be described by selected rock uniaxial compressive strength Rc.

(4) Development degree of surrounding rock joints. Joints are small fracture structures with no significant displacement on both sides of rock mass after stress fracture. The greater the degree of joint development, the greater the thickness of loosening zone.

(5) Sectional area of roadway. The size of roadway section also makes a difference to the thickness of loose zone which is proportional to the roadway section area.

## 4.2. Sample Selection

The surrounding rock loose zones of some mines in China have been counted [66,67]. The statistical data is shown in Table 1. Five indexes affecting the thickness of surrounding rock loose zone are taken as input vectors, and the thickness of surrounding rock loose zone is taken as output vectors. A prediction model of surrounding rock loosening zone based on WRVM has been established. Sample sets 1–40 of data were selected as training data, and sample sets 41–55 were used as prediction data.

**Table 1.** Statistics of surrounding rock loose zones in mines.

| N | Input | | | | | Output |
|---|---|---|---|---|---|---|
| | Depth (m) | Span/(m) | Sectional Area (m$^2$) | Strength (Mpa) | Development Degree | Thickness (m) |
| 1 | 362 | 2.6 | 6.8 | 62.4 | 2 | 0.6 |
| 2 | 660 | 4.4 | 14.6 | 12.5 | 5 | 2.2 |
| 3 | 384 | 3.5 | 11.5 | 8.5 | 3 | 1.2 |
| 4 | 150 | 3.6 | 11.7 | 14.6 | 2 | 0.6 |
| 5 | 178 | 2.6 | 6.4 | 23.8 | 3 | 1.2 |
| 6 | 510 | 3.2 | 7.3 | 12.6 | 4 | 1.6 |
| 7 | 420 | 3.6 | 10.3 | 14.3 | 3 | 1.1 |
| 8 | 450 | 3.4 | 7.8 | 9.1 | 5 | 2 |
| 9 | 236 | 3 | 7.5 | 14.3 | 3 | 1.2 |
| 10 | 470 | 4 | 12.6 | 10.1 | 5 | 2.2 |
| 11 | 467 | 3.4 | 9.6 | 10.1 | 4 | 1.8 |
| 12 | 490 | 3.7 | 8.9 | 12.5 | 4 | 1.5 |
| 13 | 450 | 3.6 | 10.8 | 13.3 | 4 | 1.6 |
| 14 | 244 | 3.4 | 8.2 | 11.2 | 3 | 1 |
| 15 | 460 | 3.2 | 9.7 | 101.6 | 1 | 0.4 |
| 16 | 373 | 2.5 | 6.3 | 14.6 | 2 | 0.9 |
| 17 | 310 | 2.8 | 7.1 | 13.8 | 3 | 1.2 |
| 18 | 125 | 2.8 | 7.1 | 13.3 | 2 | 0.7 |
| 19 | 392 | 2.8 | 6.9 | 14.5 | 2 | 0.8 |
| 20 | 249 | 3.4 | 8.2 | 16.8 | 3 | 1 |
| 21 | 140 | 3.6 | 10.3 | 13.4 | 2 | 0.5 |
| 22 | 345 | 3 | 7.6 | 65 | 2 | 0.7 |
| 23 | 315 | 2.8 | 7.1 | 11.2 | 3 | 1.1 |
| 24 | 550 | 3.4 | 9.4 | 12.5 | 5 | 2.1 |
| 25 | 410 | 3.2 | 7.2 | 13.3 | 3 | 1.1 |
| 26 | 420 | 3.2 | 9.2 | 9.1 | 4 | 1.7 |
| 27 | 340 | 3.2 | 9.2 | 19.8 | 3 | 1.3 |
| 28 | 340 | 3.2 | 9.6 | 32.2 | 2 | 0.7 |
| 29 | 420 | 3.7 | 8.9 | 9.1 | 4 | 1.4 |
| 30 | 370 | 3.5 | 8.3 | 10.5 | 3 | 1 |
| 31 | 428 | 3.6 | 11.7 | 16.5 | 3 | 1.2 |
| 32 | 465 | 4 | 12.6 | 9.5 | 4 | 1.6 |
| 33 | 403 | 2.9 | 7.2 | 12.6 | 3 | 1.3 |
| 34 | 689 | 3 | 7.6 | 15.1 | 4 | 1.8 |
| 35 | 450 | 3 | 7.6 | 11.2 | 3 | 1.2 |
| 36 | 410 | 3.6 | 11.7 | 13.3 | 4 | 1.4 |
| 37 | 348 | 3.2 | 9.2 | 7.5 | 3 | 1.2 |
| 38 | 357 | 3.2 | 8.5 | 10.5 | 3 | 1.1 |
| 39 | 276 | 2.6 | 6.6 | 15.9 | 2 | 0.8 |
| 40 | 280 | 2.8 | 7.1 | 12.7 | 2 | 0.8 |
| 41 | 321 | 2.6 | 6.6 | 13.3 | 3 | 1.1 |
| 42 | 665 | 4.4 | 14.6 | 10.9 | 4 | 1.7 |
| 43 | 350 | 3.2 | 8.5 | 10.5 | 3 | 1.2 |
| 44 | 321 | 2.6 | 6.6 | 9.2 | 3 | 1.2 |
| 45 | 340 | 3 | 7.6 | 73.6 | 2 | 0.8 |
| 46 | 470 | 3.6 | 11.2 | 9.1 | 5 | 2.1 |
| 47 | 231 | 3 | 7.5 | 18.3 | 2 | 0.7 |
| 48 | 125 | 3.4 | 9.8 | 13.3 | 3 | 1 |
| 49 | 296 | 3.4 | 7.8 | 22.4 | 4 | 1.4 |
| 50 | 435 | 2.8 | 7.2 | 15.2 | 3 | 1.2 |
| 51 | 343 | 3.2 | 9.6 | 32.2 | 2 | 0.7 |
| 52 | 525 | 3.2 | 7.3 | 15.8 | 4 | 1.6 |
| 53 | 264 | 3.2 | 9.2 | 11.2 | 3 | 1.1 |
| 54 | 292 | 3.4 | 7.8 | 12.5 | 4 | 1.4 |
| 55 | 362 | 2.6 | 6.8 | 58 | 2 | 0.8 |

Note: The development degree of surrounding rock joints is represented by 1, 2, 3, 4 and 5 joints which are very underdeveloped, underdeveloped, moderately developed, relatively developed and developed, respectively.

### 4.3. Kernel Function Parameter Determination

Cross-validation is a commonly used method to select the optimal parameters. By using cross-validation method, 28 groups of training set were randomly selected in the first 40 groups, and the remaining 12 groups were used as the validation set. The model was established by setting different parameters of the kernel function to validate the kernel function parameters. Through six cross validations, the RMS error of validation set under different kernel parameters is shown in Figure 4. According to the results of six cross validations, the optimal value of the model core parameters is chosen when the root mean square error is the smallest. According to the calculation results, the kernel parameters of Gauss-RVM are 12.5, and those of the Mexican relevance vector machine, the Morlet relevance vector machine, the and DOG relevance vector machine are 20, 22.5 and 12.5, respectively.

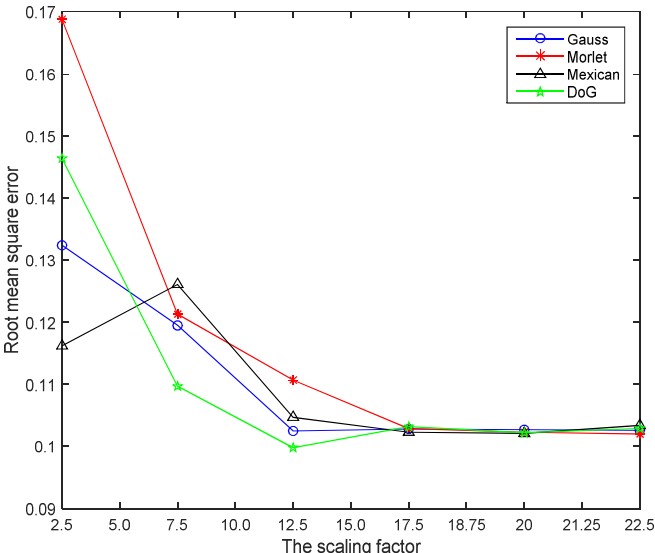

**Figure 4.** Root mean square (RMS) errors of verification sets under different kernel parameter values.

The experimental results of root mean square (RMS) errors of support vector machine (SVM), relevance vector machine (Gauss-RVM), wavelet relevance vector machine (Mexican relevance vector machine, Morlet relevance vector machine and DOG relevance vector machine) are compared, as illustrated in Table 2.

**Table 2.** Comparison of RMS errors.

| Model | SVM | Gauss-RVM | Mexican-RVM | Morlet-RVM | DOG-RVM |
|---|---|---|---|---|---|
| RMS error | 0.1780 | 0.1025 | 0.1020 | 0.1021 | 0.0998 |

By analyzing Table 2, all three WRVMs have good modeling robustness, and the modeling accuracy is better than the other two methods.

### 4.4. Result Analysis

Between the 41 to 55 groups of samples selected as test data, the predictive results of the loose zone can be calculated by inputting the data into the prediction model of the surrounding rock loose zone using WRVM. The average relative prediction errors of support vector machine (SVM), relevance vector machine (Gauss-RVM), wavelet relevance vector machine (Mexican relevance vector machine, Morlet relevance vector machine and DOG relevance vector machine) models are illustrated in Table 3.

**Table 3.** Average relative prediction errors of each model.

| Model | SVM | Gauss-RVM | Mexican-RVM | Morlet-RVM | DOG-RVM |
|---|---|---|---|---|---|
| Average relative prediction errors | 8.739% | 4.624% | 4.581% | 4.586% | 4.575% |

Table 3 shows that the average relative prediction errors of the Mexican relevance vector machine, the Morlet relevance vector machine, and the DOG relevance vector machine models are accordingly 4.581%, 4.586% and 4.575%. WRVM has higher accuracy than SVM and RVM. It shows that the model has achieved some accuracy and reliability in practical engineering applications. Taking the Mexican-RVM forecasting model as an example, the results of training sample modeling were analyzed as illustrated in Figure 5, the model forecasting results in Figure 6, model forecasting results in Figure 7, and model forecasting error results in Figure 8.

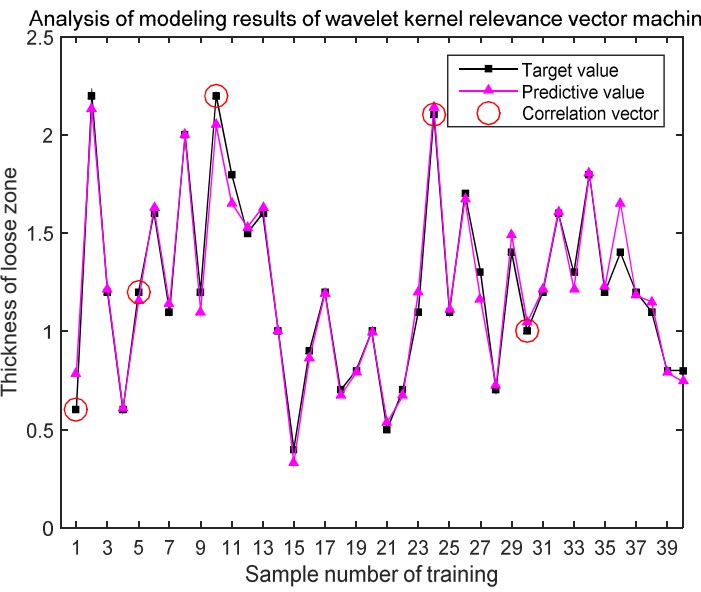

**Figure 5.** Analysis of modeling results of wavelet relevance vector machine (WRVM).

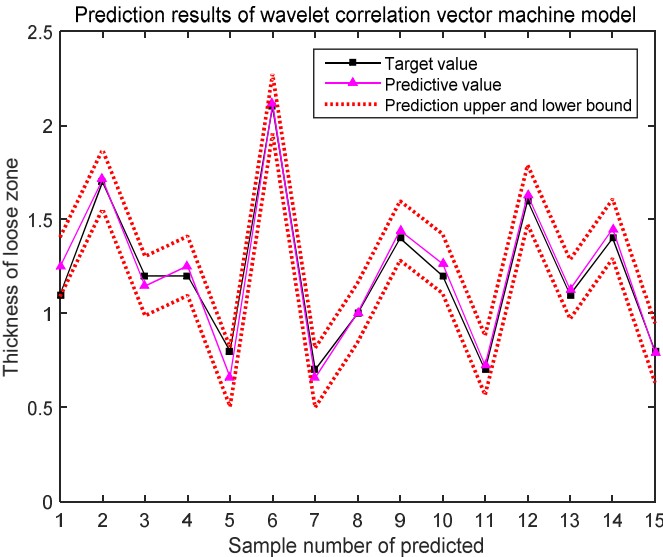

**Figure 6.** Prediction results of WRVM model.

Figure 5 shows that the prediction model of WRVM trained with 40 sets of data has a good fitting effect. There are five relevance vectors. The number of relevance vectors is reasonable. They reflect the core characteristics of the data.

Figure 6 shows the predicted sample's target value and predictive value and prediction upper and lower bound and also gives the interval information of the predicted value based on probability, which increases the reliability of the predicted results and provides more decision support for the prediction of the surrounding rock loosening zone in engineering practice.

The paper further analyzed the fitting effect between the predicted results and the real values. The complex relevance coefficient R of the predicted samples is 0.97613, and the fitting effect is shown in Figure 7. The square correlation coefficient is 0.95 > 0.9, which further demonstrates that the fitting results of the prediction results are better. From Figure 8, it can be seen that only two roadways have a prediction error of more than 10%. In general, the predicted result has high reliability.

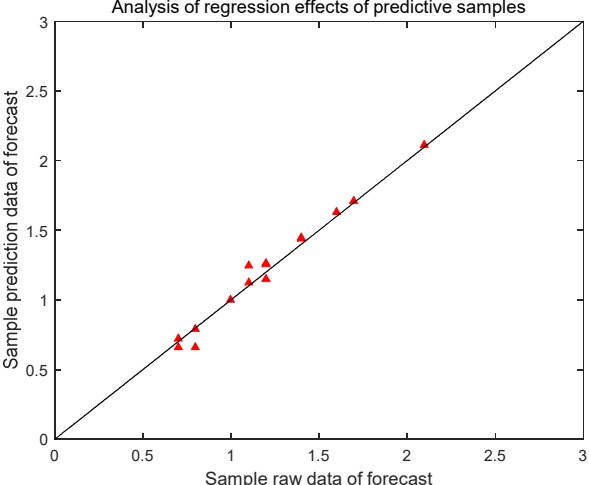

**Figure 7.** Analysis of the fitting effect.

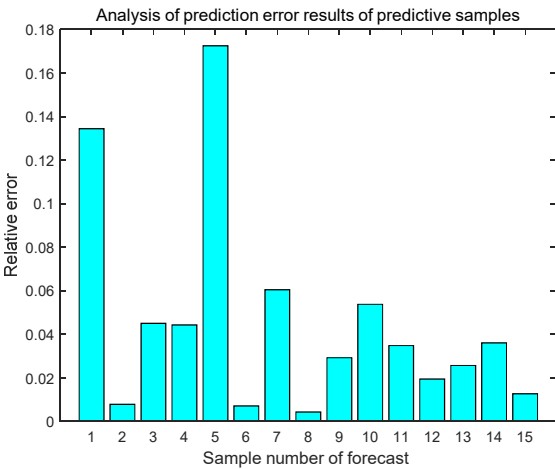

**Figure 8.** Prediction error of prediction samples.

The results of SVM prediction model are shown in Figure 9, the results of relevance vector machine prediction model are shown in Figure 10, and the calculation efficiency of each prediction model is shown in Table 4. Through further comparative analysis, it is found that the accuracy and reliability of the wavelet correlation vector machine prediction model are better.

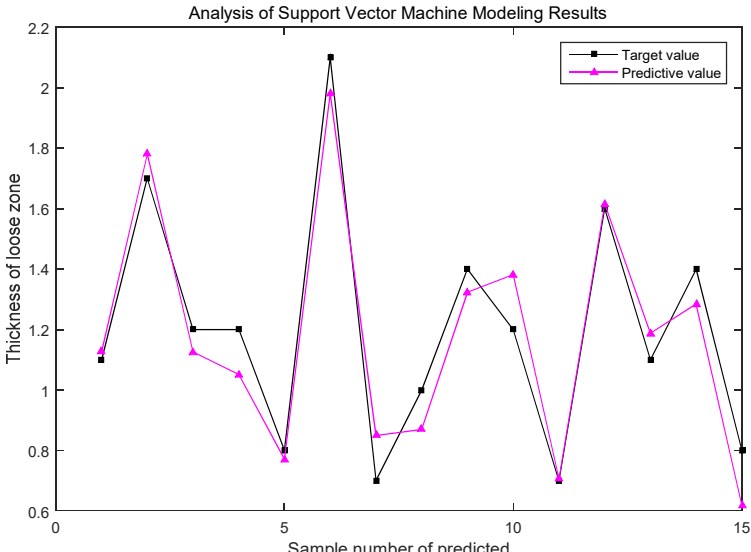

**Figure 9.** Prediction results of SVM model.

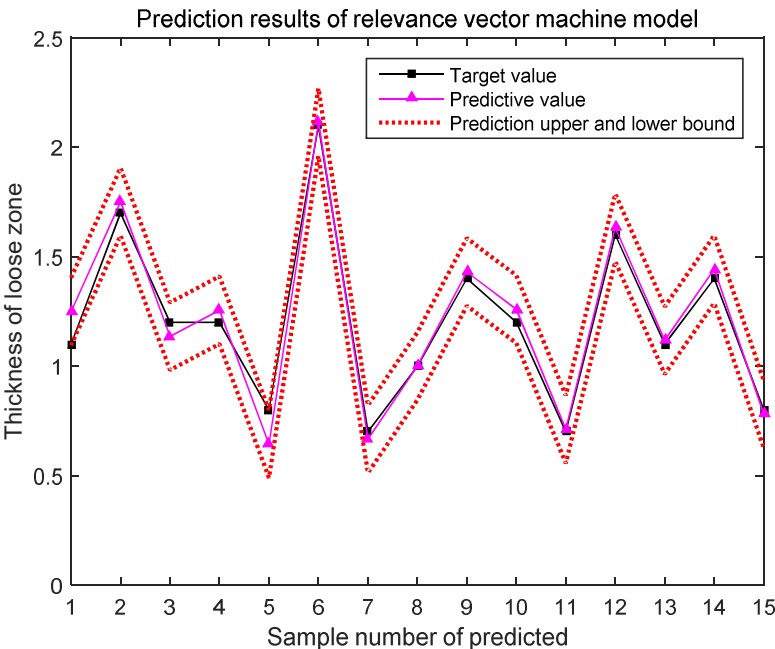

**Figure 10.** Prediction results of RVM model.

**Table 4.** Computational efficiency of each model.

| Model | SVM | RVM | WRVM |
|---|---|---|---|
| Running time | 3.729 s | 1.435 s | 0.776 s |

## 5. Conclusions

Based on the theory of RVM, the wavelet function was introduced to replace the original Gauss function as the model kernel function. Five factors affecting the loose zone of roadway surrounding rock have been selected as the model input. The predictive model of the loose zone of roadway surrounding rock using WRVM was established. Mean square error, relative prediction error and square correlation coefficient were introduced to estimate the model accuracy. The major findings for the paper are listed below.

(1) By applying cross-validation, the kernel parameters of were respectively calculated to be 20, 22.5 and 12.5. By comparing the RMS error of the experimental results of SVM, Gauss-RVM and WRVM, the advantages and accuracy of the proposed WRVM model are verified.

(2) In practical engineering applications, the average relative prediction errors of three WRVMs models are accordingly 4.581%, 4.586% and 4.575%. The probability intervals of prediction values were given. The square correlation coefficient of prediction samples is 0.95 > 0.9, which further verifies the accuracy and reliability of the predictive model of roadway surrounding rock loose zone using the proposed WRVM.

**Author Contributions:** Y.L. contributed to the analytical method and program codes, taking charge of data analysis and paper writing while Q.W. and Y.Y. were responsible for the research supervision and paper revising. X.L. and W.W. took charge of the data analysis. The last script has been perused and approved by all writers.

**Funding:** The research got the support of the National Natural Science Foundation of China (No. 51574183) and Open Fund of Hubei Key Laboratory of Mechanical Transmission and Manufacturing Engineering of China (No. 2018A09).

**Conflicts of Interest:** The authors declare no conflicts of interest.

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
