# Peer review of "Predicting the Loose Zone of Roadway Surrounding Rock Using Wavelet Relevance Vector Machine"

_applsci, doi:10.3390/app9102064_

Reviewer 1 Report

In this manuscripts, based on the theory of relevance vector machine (RVM), the wavelet function was introduced to replace the original Gauss function as the model kernel function to form the WRVM. Five factors affecting the loose zone of roadway surrounding rock were selected as the model input, and the prediction model of the loose zone of roadway surrounding rock based on wavelet relevance vector machine (WRVM) was established. By using cross-validation method, the kernel parameters of three kinds of wavelet correlation vector machines (RVMs) were calculated. By comparing and analyzing the root mean square (RMS) error of the test results of each predictive model, the advantages and accuracy of the model were verified. In practical engineering applications, Morlet relevance vector machine and DOG relevance vector 22 machine models were accordingly 4.581%, 4.586%, and 4.575%. The square correlation coefficient of the predicted samples was 0.95 > 0.90, which further verified the accuracy and reliability of the proposed method.     The subject addressed is within the scope of the journal. However, the manuscript in the present version contains several weaknesses. Appropriate revisions should be undertaken in order to justify recommendation for publication.    1. It is mentioned that relevance vector machine (RVM) was used to predict the loose zone of roadway surrounding rock. What are the advantages of adopting this particular soft computing technique over others (e.g., SVM, RT, MLP, GP, and MT etc.) in this case? How will this affect the results? More details should be furnished.    2. A wavelet decomposition (WD), one of data pre-processing methods, is required to increase the model accuracy using decomposing original input and target time series into sub-time series. Only a relevance vector machine model cannot achieve the reliable prediction because of high nonlinearity for the loose zone of roadway surrounding rock. More details should be furnished.    3. It is mentioned that three quantitative performance indices are adopted to assess the model performances. What are the other feasible alternatives? What are the advantages of adopting these particular performance indices over others in this case? How will this affect the results? More details should be furnished.    4. Some key parameters of relevance vector machine are not mentioned. The rationale on the choice of the particular set of parameters should be explained with more details. Have the authors experimented with other sets of values? What are the sensitivities of these parameters on the results?    5. The discussion section in the present version is relatively weak and should be strengthened with more details and justifications.    6. There are some occasional grammatical problems within the text (e.g., abstract, introduction, and conclusions etc.). It may need the attention of someone fluent in English language to enhance the readability.   7. Since all figures have low resolution printing, the reviewer cannot recognize them clearly. Please revise them with high resolution.   8. The authors have to add the state-of-the art references in the manuscripts. Relevant researches within worldwide schemes can be found from many journals.

Author Response

Detailed Response to the reviewers’ comments

Journal: Applied Sciences

Manuscript Number: applsci-484811

Title: " Predicting the Loose Zone of Roadway Surrounding Rock Using Wavelet Relevance Vector Machine "

Author(s): Yang Liu, Yicheng Ye, Qihu Wang, Xiaoyun Liu, Weiqi Wang,

 Dear Reviewer:

We would like to thank the reviewers for their valuable comments, which have greatly helped to improve the quality of the paper. We have made significant changes on the manuscript as suggested by the reviewers (the main changes are highlighted in red in the revised manuscript). We hope that our answers are satisfactory. Please don’t hesitate to contact me if you have any questions.

Comment #1. It is mentioned that relevance vector machine (RVM) was used to predict the loose zone of roadway surrounding rock. What are the advantages of adopting this particular soft computing technique over others (e.g., SVM, RT, MLP, GP, and MT etc.) in this case? How will this affect the results? More details should be furnished.

Answer #1: Thank you for your suggestion. We have made some modifications. Based on the support vector machine (SVM) and the engineering background, the method of using correlation vector machine (CVM) is proposed, which adds the difference between correlation vector machine (CVM) and support vector machine (SVM) and the advantages of correlation vector machine (SVM). RVM can be used for regression estimation and prediction to obtain the distribution of predicted values. It not only has excellent non-linear fitting and generalization ability, but also makes up for the defect that SVM needs to estimate regularization parameters and cannot get the prediction results based on probability.

Comment #2. A wavelet decomposition (WD), one of data pre-processing methods, is required to increase the model accuracy using decomposing original input and target time series into sub-time series. Only a relevance vector machine model cannot achieve the reliable prediction because of high nonlinearity for the loose zone of roadway surrounding rock. More details should be furnished.

Answer #2: Thank you for your suggestion. We have made some modifications. Because of the non-linear correlation between the factors affecting the surrounding rock loosening zone and the uncertainty of its state and parameters measurement, the prediction of the surrounding rock loosening zone is a typical random and fuzzy problem. The predictive method of correlation vector machine can solve the practical problems of non-linearity and small sample, and has achieved certain results in engineering application. Wavelet is a function approximation tool in square integrable space. The wavelet kernel function is applied to support vector machine function approximation, which effectively improves the accuracy of the model. Wavelet technology has great potential in dealing with the approximation and classification of non-stationary data. A wavelet function is a set of bases that can approximate any function. Therefore, this paper combines wavelet technology with RVM, namely wavelet correlation vector machine (WRVM), and applies it to the prediction of surrounding rock loose zone of roadway, which can effectively solve the prediction problem of randomness and ambiguity. It has been revised and supplemented in the text.

Comment #3. It is mentioned that three quantitative performance indices are adopted to assess the model performances. What are the other feasible alternatives? What are the advantages of adopting these particular performance indices over others in this case? How will this affect the results? More details should be furnished.

Answer #3: Thank you for your suggestion. RMSE is the square root of the ratio of the square of the predicted value to the true value and the number n of observations. In the actual measurement, the number of observations n is always finite, and the true value can only be replaced by the most reliable value. The RMSE is very sensitive to very large or very small errors in a set of measurements, so the RMSE is a good reflection of the precision of the measurement. RPE is the percentage of the absolute gap relative to the observed value. The prediction error is determined to test the accuracy of the prediction and provide a reliable basis for decision making. Test of the degree of fit between R2 and the sample regression line and sample observations. The PRE results of the five methods are shown in Table 3. Figure 7 shows that the complex correlation coefficient r between the predicted results and the actual values is 0.97613.

Comment #4. Some key parameters of relevance vector machine are not mentioned. The rationale on the choice of the particular set of parameters should be explained with more details. Have the authors experimented with other sets of values? What are the sensitivities of these parameters on the results?

Answer #4: Thank you for your suggestion. We have made some modifications. The scaling factor of the wavelet kernel function is a constant that needs to be optimized. The expansion factor is selected by cross validation method according to figure 4. The calculation accuracy of the model is improved.

Comment #5. The discussion section in the present version is relatively weak and should be strengthened with more details and justifications.

Answer #5: Thank you for your suggestion. We have made some modifications. In the future, we will carry out further research on the algorithm.

Comment #6. There are some occasional grammatical problems within the text (e.g., abstract, introduction, and conclusions etc.). It may need the attention of someone fluent in English language to enhance the readability.

Answer #6: Thank you for your suggestion. We have made some modifications.

Comment #7. Since all figures have low resolution printing, the reviewer cannot recognize them clearly. Please revise them with high resolution.

Answer #7: Thank you for your suggestion. We have made some modifications.

Comment #8. The authors have to add the state-of-the art references in the manuscripts. Relevant researches within worldwide schemes can be found from many journals.

Answer #8: Thank you for your suggestion. We have made some modifications.

Reviewer 2 Report

The introduction is clear and well referenced. Authors must clarify the novelty of the research in the text. The application of  Wavelet to Relevant Vector Machine is not a novelty.  Results can be improved and extended.

Some comments:

-Pag 3, lines 96-97 are the same as lines 98-99. Please check and delete lines 96-97.

-Please, check the format of the symbols on the text, i.e., pag 4, lines 131-132.

-The section 2 is very hard to understand, in my opinion, too many formulas that can be found in other references. The results section do not show the information explained.

-Please, explain the reasons of choosing those 3 kinds of wavelet (section 2.2).

-Figure 2 is very explanatory, consider the option of making  a similar figure for section 2.1. It could be improve with the results of each step

- Some explanation about SVM could be added before 2.1.

-Check the meaning of "concrete"  in page 6, line 160

- Change the sentence: "Five indexes, including roadway burial depth, roadway span, surrounding rock strength (uniaxial compressive strength), development degree of surrounding rock joints and roadway cross-section area, are used to predict the thickness of loose zone." by "Five indexes are used to predict the thickness of loose zone:"

-Table 1,use the same number of decimal for the same parameters. Add a row to "Input" and "Output". Indicate the units in parentheses, i.e., Span (m)

-Figure 4. X axis, use an appropiate title. Figure caption: do not use a capital letter in each Word.

-Please, explain the last sentence before Figure 4, why the kernel parameters are the indicated values? Not only "According to the calculation results"

-Once the meaning of RMS, RVM, WRVM have been explained, it is not necessary to do it again, for example, first sentence of page 11 and section 4.4.

-The results of Tables 3 and 4 shown very similar values for Gauss-RVM and WRVM. The authors said that "the modeling accuracy is obviously better than the other two methods.", the Word "obviously" is not the more appropiate.

- For a better evaluation of the results it would be necessary to know information about the calculation time or computational resources needed to perform the different calculations, both RVM and WRVM. Could the authors provide information about it? If the calculation of WRVM need much more computational resources than RVM calculations, maybe the RVM results are acceptable.

-Line 250, check that figures 6 and 7 shows.

-Figure 5 and 6, use the same Y-axis scale.

-There is not practical information about the algorithm procedure. How many iterations were neccesary? The theoretical section are very extense but the results are very brief. Results section can be improved and extended.

-3 errors are explained in section 2.4, but only the RMS results are shown in the paper. Please, shows the RPE and R^2 results or delete from the text and the conclusions. What is the average relative prediction errors of section 4.4?

-Is it possible to apply this algorithm using a different number of input? i.e,, 4 or 7 indices?

-Is it possible to apply this algorithm to other applications? 

Author Response

Detailed Response to the reviewers’ comments

Journal: Applied Sciences

Manuscript Number: applsci-484811

Title: " Predicting the Loose Zone of Roadway Surrounding Rock Using Wavelet Relevance Vector Machine "

Author(s): Yang Liu, Yicheng Ye, Qihu Wang, Xiaoyun Liu, Weiqi Wang,

 Dear Reviewer:

We would like to thank the reviewers for their valuable comments, which have greatly helped to improve the quality of the paper. We have made significant changes on the manuscript as suggested by the reviewers (the main changes are highlighted in red in the revised manuscript). We hope that our answers are satisfactory. Please don’t hesitate to contact me if you have any questions.

Comment #1. -Pag 3, lines 96-97 are the same as lines 98-99. Please check and delete lines 96-97.

Answer #1: Thank you for your suggestion. We have deleted the same content and modified it.

Comment #2. -Please, check the format of the symbols on the text, i.e., pag 4, lines 131-132.

Answer #2: Thank you for your suggestion. We have checked and revised it.

Comment #3. -The section 2 is very hard to understand, in my opinion, too many formulas that can be found in other references. The results section do not show the information explained.

Answer #3: Thank you for your suggestion. We have made some adjustments, because many of them are the theoretical basis of the correlation vector machine.

Comment #4. -Please, explain the reasons of choosing those 3 kinds of wavelet (section 2.2).

Answer #4: Thank you for your suggestion. In order to illustrate the universality of the wavelet correlation vector machine, we choose three commonly used wavelet kernel functions. These three kinds of wavelet cores have their own characteristics, which can reflect the advantages of constructing translation invariant wavelet cores from different angles.

Comment #5. -Figure 2 is very explanatory, consider the option of making  a similar figure for section 2.1. It could be improve with the results of each step.

Answer #5: Thank you for your suggestion. The steps shown in Figure 2, which include sections 2.1 and 2.2, have been revised in accordance with your suggestions.

Comment #6. Some explanation about SVM could be added before 2.1.

Answer #6: Thank you for your suggestion. We have already made supplements and explanations.

Comment #7. Check the meaning of "concrete"  in page 6, line 160.

Answer #7: Thank you for your suggestion. We have made some modifications.

Comment #8. Change the sentence: "Five indexes, including roadway burial depth, roadway span, surrounding rock strength (uniaxial compressive strength), development degree of surrounding rock joints and roadway cross-section area, are used to predict the thickness of loose zone." by "Five indexes are used to predict the thickness of loose zone:".

Answer #8: Thank you for your suggestion. We have made some modifications.

Comment #9. Table 1,use the same number of decimal for the same parameters. Add a row to "Input" and "Output". Indicate the units in parentheses, i.e., Span (m).

Answer #9: Thank you for your suggestion. In the first step, we normalize the data and modify the tables accordingly.

Comment #10. Figure 4. X axis, use an appropiate title. Figure caption: do not use a capital letter in each Word.

Answer #10: Thank you for your suggestion. We have made some modifications.

Comment #11. Please, explain the last sentence before Figure 4, why the kernel parameters are the indicated values? Not only "According to the calculation results".

Answer #11: Thank you for your suggestion. We have made some modifications. Cross-validation is a commonly used method to select the optimal parameters. According to the results of six cross validations, the optimal value of the model core parameters is chosen when the root mean square error is the smallest.

Comment #12. Once the meaning of RMS, RVM, WRVM have been explained, it is not necessary to do it again, for example, first sentence of page 11 and section 4.4.

Answer #12: Thank you for your suggestion. We have made some modifications.

Comment #13. The results of Tables 3 and 4 shown very similar values for Gauss-RVM and WRVM. The authors said that "the modeling accuracy is obviously better than the other two methods.", the Word "obviously" is not the more appropiate.

Answer #13: Thank you for your suggestion. We have made some modifications.

Comment #14. For a better evaluation of the results it would be necessary to know information about the calculation time or computational resources needed to perform the different calculations, both RVM and WRVM. Could the authors provide information about it? If the calculation of WRVM need much more computational resources than RVM calculations, maybe the RVM results are acceptable.

Answer #14: Thank you for your suggestion. Because the total sample size is small, the calculation time of RVM and WRVM is almost the same, so the selection accuracy is compared, and the comparison of time and resources is not given.

Comment #15. Line 250, check that figures 6 and 7 shows.

Answer #15: Thank you for your suggestion. We have checked and adjusted it.

Comment #16. Figure 5 and 6, use the same Y-axis scale.

Answer #16: Thank you for your suggestion. We have made some modifications.

Comment #17. There is not practical information about the algorithm procedure. How many iterations were neccesary? The theoretical section are very extense but the results are very brief. Results section can be improved and extended.

Answer #17: Thank you for your suggestion. In the conclusion, the accuracy of the five methods is compared. The paper focuses on the application of model prediction method in the actual engineering background. In the later stage, the results will be improved and expanded from the perspective of algorithm.

Comment #18. 3 errors are explained in section 2.4, but only the RMS results are shown in the paper. Please, shows the RPE and R^2 results or delete from the text and the conclusions. What is the average relative prediction errors of section 4.4?

Answer #18: Thank you for your suggestion. The PRE results of the five methods are shown in Table 3. Figure 7 shows that the complex correlation coefficient r between the predicted results and the actual values is 0.97613.

Comment #19. Is it possible to apply this algorithm using a different number of input? i.e,, 4 or 7 indices?

Answer #19: Thank you for your suggestion. Based on the relevant literature [46, 64-65] and the principles of simplicity, independence and accessibility, the influence of in-situ stress on the size of loose zone is reflected by roadway burial depth, and the influence of roadway span and roadway section area on the size of loose zone is reflected by the geometric size and shape of roadway. When the number of input indicators is 4 or 7, the model is applicable, but the indicators should be selected according to the actual characteristics of the project.

Comment #20. Is it possible to apply this algorithm to other applications?

Answer #20: Thank you for your suggestion. Yes, the algorithm has been applied in product quality model and so on [49].

Round  2

Reviewer 1 Report

The authors carried out the revision of reviewed manuscripts which the reviewer wants to check carefully. However, the response for comment 5 cannot satisfy reviewer’s intention. Please add or modify the discussion section.

Author Response

Detailed Response to the reviewers’ comments

Journal: Applied Sciences

Manuscript Number: applsci-484811

Title: " Predicting the Loose Zone of Roadway Surrounding Rock Using Wavelet Relevance Vector Machine "

Author(s): Yang Liu, Yicheng Ye, Qihu Wang, Xiaoyun Liu, Weiqi Wang,

 Dear Reviewer:

We would like to thank the reviewers for their valuable comments, which have greatly helped to improve the quality of the paper. We have made significant changes on the manuscript as suggested by the reviewers (the main changes are highlighted in red in the revised manuscript). We hope that our answers are satisfactory. Please don’t hesitate to contact me if you have any questions.

Comment #5. The discussion section in the present version is relatively weak and should be strengthened with more details and justifications.

Answer #5: Thank you for your suggestion. The calculation results of support vector machine prediction model and correlation vector machine prediction model have been supplemented. Computational efficiency comparison tables of each prediction model are added. Your suggestions are very instructive and can improve the quality of the article. For further improvement of the model, we will do further research.

Round  3

Reviewer 1 Report

The authors carried out the revision of reviewed manuscripts which the reviewer wants to check carefully. So, I think that the manuscripts have to be accepted for the publication.